# Fabrication and Characterization of Gelatin/Zein Nanofiber Films Loading Perillaldehyde for the Preservation of Chilled Chicken

**DOI:** 10.3390/foods10061277

**Published:** 2021-06-03

**Authors:** Debao Wang, Yini Liu, Jinyue Sun, Zhilan Sun, Fang Liu, Lihui Du, Daoying Wang

**Affiliations:** 1Institute of Agricultural Products Processing, Jiangsu Academy of Agricultural Sciences, Nanjing 210014, China; Debaowang_2021@163.com (D.W.); yini_l@163.com (Y.L.); 15247189175@163.com (J.S.); sunzhilan@jaas.ac.cn (Z.S.); wangdaoying@jaas.ac.cn (D.W.); 2School of Food and Biological Engineering, Jiangsu University, Zhenjiang 212013, China; 3Collaborative Innovation Center for Modern Grain Circulation and Safety, Key Laboratory of Grains and Oils Quality Control and Processing, College of Food Science and Engineering, Nanjing University of Finance and Economics, Nanjing 210023, China

**Keywords:** perillaldehyde, nanofiber, zein, antibacterial activity, preservation

## Abstract

Perillaldehyde is a natural antibacterial agent extracted from perilla essential oil. In our methodology, five antibacterial nanofiber packaging films are prepared by loading different concentrations of perillaldehyde (P) into gelatin/zein (G/Z) polymers. Morphology observations show that the G/Z/P film had a good uniform microstructure and nano-diameter as the weight ratio of 5:1:0.02 (G/Z/P). Fourier transform infrared spectroscopy, and X-ray indicate that these three ingredients had good compatibility and strong interaction via hydrogen bonding. Water contact angle results show that the G/Z/P films gradually change from hydrophilic to hydrophobic with the increase of perillaldehyde. Thermal analysis indicates that the G/Z/P (5:1:0.02) film has good thermal stability. Antibacterial and storage analysis indicates that G/Z/P (5:1:0.02) film is effective to inactivate *Staphylococcus aureus* and *Salmonella enteritidis*, and obviously reduces the increasing rate of total bacteria counts and volatile basic nitrogen of chicken breasts. This study indicates that the G/Z/P (5:1:0.02) is a kind of potential antibacterial food packaging film.

## 1. Introduction

People are gradually pursuing health, nutrition, and diversification of diet, especially fresh food, such as fruits, vegetables, meat, and aquatic products, as living standards improve [1]. The freshness of foods is a key indicator for consumers. Fresh food faces corruption and deterioration caused by microorganisms in the process of refrigeration and sales, which results in a short shelf life [2,3,4]. Developing antibacterial packaging films is crucial to ensure the quality of fresh food and longer shelf life [1,5]. Antibacterial packaging is an emerging technology that combines packaging films and antibacterial agents to inhibit the growth activity of spoilage bacteria and extend the shelf life of food [6,7]. In recent years, many reports focused on preparing nanofiber antibacterial films by loading natural antibacterial agents on food-grade biopolymers using electrospinning technology [5,8,9]. Electrospinning technology involves a polymer solution, which generates a charged jet by overcoming surface tension under the action of a high-voltage electrostatic field and finally solidifies to obtain nanofibers [10]. Nanofibers and electrospinning technology are favored in the food packaging and medical fields because of the structural integrity, high surface area, and volume of the electrospinning fiber and the particularity of the fiber arrangement [11,12]. Nanofibers used in the food package and other food fields need to be prepared entirely from food biopolymers, such as nano cellulose, gelatin, zein, and alginate so on, which are due to their biodegradable, renewable, and safety characteristics [13].

Gelatin, one of the most commonly used natural protein-based biopolymer approved by FDA-approved, is derived from collagen in animal bones, connective tissue, and skin [14]. Gelatin is recognized as a potential candidate for nanofiber synthesis because of its biodegradability, bioactivity, and nontoxicity [15]. In the field of food packaging, gelatin is often used to prepare nanofiber membranes with other polymers. Similar to gelatin, zein is also used in many food applications [5,16]. Zein, a kind of food protein widely found in corn, is only soluble in certain organic solvents because three-quarters of its amino acid residues are hydrophobic, and one-quarter is hydrophilic [17]. Zein can bind to gelatin through strong hydrogen bonds to maintain a stable nanofiber film structure [18]. Compared with pure gelatin or zein nanofibers, gelatin/zein (G/Z) nanofibers have good deformation and flexibility and still maintain a 3-D porous structure after soaking in water or ethanol for 24 h [19,20].

Nanofiber films are frequently used in the food industry as carriers for encapsulation of biologically active antibacterial substances, such as ε-polylysine [19], curcumin [9], cinnamic aldehyde [21], and essential oils [20], to inhibit the activity of spoilage bacteria and pathogenic bacteria and extend the shelf life of the preservation of meat products. According to Catto et al. [22], perilla essential oil has effective antibacterial properties. Research on the volatile essential oil components of Perilla frutescens indicates that this plant is an especially good source of perillaldehyde and has the highest perillaldehyde content [23]. Perillaldehyde can cause cell damage, which causes the leakage of large intracellular molecules and cell death [24,25]. However, few studies have reported the effect of the incorporation of perillaldehyde in electrospun nanofiber on appearance characteristics and antibacterial activity.

In this paper, different antibacterial nanofiber films were prepared by electrospinning using perillaldehyde as the loaded antibacterial agent and gelatin and zein as the carriers. The morphology, diameter distribution, porosity, molecular interaction, and thermal stability of the G/Z and G/Z/P films were characterized. The antibacterial activities of the nanofiber films were investigated using foodborne pathogens of *Staphylococcus aureus* and *Salmonella enteritidis*. Finally, the effect of the G/Z/P (5:1:0.02) on the preservation of chilled chicken was explored.

## 2. Materials and Methods

### 2.1. Chemicals

Zein from maize (biological reagent, 92%) was purchased from Yifeixue Biological Technology Co., Ltd. (Nanjing, China). Gelatin and perillaldehyde (biological reagent, ≥92%) were obtained from Yuanye Biological Technology Co., Ltd. (Shanghai, China).

### 2.2. Preparation of Electrospinning Solutions

Gelatin (12.5% *w/v*) was dissolved in acetic acid solution (80% *v/v* in distilled water) and stirred on a magnetic stirrer for 15 min at 50 °C to obtain a clear and homogenous solution. Zein was added to the solution to form a final concentration of 2.5% (*w/v*). The G/Z solution was mechanically stirred at 50 °C for 15–30 min to achieve complete dissolution. Different concentrations of perillaldehyde (62.25, 125, 250, 500, and 1000 µg mL^−1^) were added to the G/Z solution and stirred for 24 h at 20–25 °C. The prepared films were designated as G/Z, G/Z/P (5:1:0.0025), G/Z/P (5:1:0.005), G/Z/P (5:1:0.01), G/Z/P (5:1:0.02), G/Z/P (5:1:0.04), respectively.

### 2.3. Properties of Electrospinning Solution

The viscosity of the mixed solutions was detected according to the method of Liu et al. [19]. The rheological properties of electrospinning solution solutions were characterized using the rotational rheometer (HAAKE MARS iQ, Thermo Fisher Scientific, Waltham, MA, USA). The values of the shear rate range from 0 to 100 s^−^^1^ were used as the solution viscosity.

### 2.4. Preparation of G/Z and G/Z/P Nanofiber Films

The injector loaded with 10 mL polymer solution was equipped with an 18-gauge steel needle and was pumped at a flow rate of 0.03 mm min^−1^. A voltage of 22 kV was applied, and the distance of the collector was kept at 10 cm. Sprayed nanofiber yarn was collected on a 25 × 25 cm aluminum foil attached to the collector tablet. Electrospinning was performed at room temperature.

### 2.5. Evaluation of Composite Nanofiber Films Characterization

#### 2.5.1. Morphology of Nanofiber Films

The nanofiber morphology of the prepared G/Z and G/Z/P films was observed using SEM (EVO-LS10, Carl Zeiss, Oberkochen, Germany). The diameter distributions were determined by measuring 100 fibers for each film sample using the ImageJ software.

#### 2.5.2. Porosity

The nanofiber films were cut into 3 × 3 cm^2^ shapes and weighed using a ten-thousandth balance (ME204E, Mettler Toledo, Switzerland). The thickness of film was measured using a vernier caliper at four different places. Apparent density (ρ_s_) was determined by the ratio of the weight and average thickness of the nanofiber films. The density (ρ_m_) of the nanofiber films was determined based on their mass percentage compositions and densities of gelatin (1.41 g cm^−3^), zein (1.22 g cm^−3^), and perillaldehyde (0.97 g cm^−3^) according to the study of Laha et al. [26]. Porosity was determined by the following formula:$$\mathrm{porosity}\left(\%\right)=\left[1-\left(\frac{\rho_{s}}{\rho_{m}}\right)\right]\times 100\%$$

#### 2.5.3. FTIR Analysis

The nanofiber films (5 mg) were analyzed by FTIR using a Nicolet iS50R instrument (Thermo Nicolet Ltd., Vernon Hills, IL, USA). The spectra were obtained at a 2 cm^−1^ resolution with 32 scans over the wavenumber range of 500–4000 cm^−1^.

#### 2.5.4. X-ray Diffraction (XRD)

The crystal structures of gelatin, zein, and their nanofibers were investigated by Explorer X-ray diffractometer using the D2 PHASER (Bruker Daltonic Inc., Germany). The XRD scanning was done at 2θ = 5°–85°, and generated at 40 kV and 30 mA.

#### 2.5.5. Thermal Analysis

Thermal stability of nanofiber films was performed by differential scanning calorimetry (DSC; Q20, TA Instruments, New Castle, DE, USA) under nitrogen atmosphere at a flow rate of 50 mL min^−1^. The samples (approximately 4 mg) were sealed in aluminum pans and heated by 10 °C min^−1^ from 25 °C to 235 °C. Thermogravimetric analysis (TGA; TA Instruments Q50, New Castle, DE, USA) was conducted under 50 mL min^−1^ nitrogen flow, and the temperature was increased by 10 °C min^−1^ from 25 °C to 600 °C.

#### 2.5.6. Water Contact Angle

The water contact angle was measured using a DSA100S drop shape analyzer (Biolin Science and Technology Co., Ltd., Gothenburg, Sweden). The nanofiber films were fixed to the table, and a droplet of ultra-pure water (3 µL) was added to the surface. The droplet was equilibrated for 3 s before measurement.

### 2.6. Antibacterial Activities of the Nanofiber Films

#### 2.6.1. Inactivation of Bacterial Cells

The effectiveness of the prepared nanofiber films in inactivating the two foodborne pathogens was further analyzed. The bacterial suspensions (approximately 10^7^ CFU mL^−1^) were divided into different tubes. One tube had 1 mL of bacterial suspension and 25 mg of nanofiber films. The samples were placed into a shaking incubator during treatment time. The bacterial suspension (1 mL) in each tube was sampled at 0, 0.5, 2, and 4 h, and the bacterial cells were counted by 10-fold dilution method and plate counting.

#### 2.6.2. Antibacterial Activity of Nanofiber Film on Chicken Breasts

Fresh chicken breast was cut in cuboids weighing 25 g and packaged in aluminum foil without nanofiber film and G/Z/P (5:1:0.02) film separately. These chicken were stored for 12 d at 4 °C. At the different sampling times, 10 g chicken sample was mixed with 90 mL sterilized normal saline in aseptic bag and homogenized for 30 min. The bacterial counts were analyzed by the 10-dilution method. Three suitable decimal dilutions were chosen and plated on plate count agar (PCA) (Shanghai Guangrui Biological Technology Co., Ltd., Shanghai, China). Each study was performed in triplicates. TVB-N was determined by the micro-diffusion method according to the standard GB 5009.228-2016 [27].

### 2.7. Statistical Analysis

Statistical analysis was performed using SPSS 19.0 (IBM, Chicago, IL, USA). Analyze the significance of the data was determined by one-way ANOVA. The graphs in the article were made with Origin 18.5 software.

## 3. Results

### 3.1. Characterization of Electrospinning Solutions

The viscosity of the solution is important for of morphology development of the nanofiber films in the electrospinning process [20]. As shown in Figure 1, the viscosity of gelatin/zein electrospinning solution loaded with perillaldehyde was less than that of the G/Z when the ratio of gelatin, zein, and perillaldehyde is 5:1:0.0025, but it was higher than that of G/Z solution when the ratio is higher than 5:1:0.005. The results indicated that the addition of perillaldehyde changed the viscosity of the electrospinning solution. The viscosity of the electrospinning solution is related to the extent of entanglement of the polymers’ side chains within the solution; the entanglement increases with the concentration of the polymer solution and results in the increase in viscosity [28,29]. A lower or higher concentration of the electrospinning copolymer will cause the polymer chain to break and form beads before reaching the collector, and only at a certain concentration will the polymer form bead-free nanofibers [14,20]. Therefore, proper solution concentration and chain entanglement are necessary for the electrospinning process.

### 3.2. Morphology and Diameter of the Nanofiber Membrane

Variations in the fiber diameter and morphology of the G/Z/P nanofibers were shown in Figure 2. Gelatin is a macromolecular hydrophilic colloid that is commonly used as a biopolymer in bioactive packaging materials after the partial hydrolysis of collagen. The nanofiber membrane made of monomer gelatin has poor water resistance and is rapidly dissolved in water [30,31]. Zein is rich in sulfur amino acids, which can bind to gelatin through strong hydrogen bonds, disulfide bonds, and hydrophobic bonds to maintain a stable nanofiber membrane structure [18]. The addition of perillaldehyde changed the structure of nanofibers. The four nanofibers of G/Z/P (5:1:0.0025), G/Z/P (5:1:0.005), G/Z/P (5:1:0.01), and G/Z/P (5:1:0.04) had many breaks and nodules (Figure 2A–C,E), while the G/Z/P (5:1:0.02) film had a similar uniform structure with the G/Z film. The mean diameter of the nanofibers increased from 52.32 nm to 70.94 nm when the perillaldehyde content in the G/Z/P solutions increased from 62.5 to 500 µg mL^−1^ (Figure 2A–E). Shao et al. [32] also reported that different loading concentrations of tea polyphenols could effectively change the diameter and morphology of pullulan carboxymethylcellulose electrospun nanofibers. The mean diameter of the nanofibers decreased from 70.94 nm to 36.63 nm when the concentration of perillaldehyde in the electrospinning solution increased from 500 to 1000 µg mL^−1^ (Figure 2D,E). This outcome may be caused by the hampered flow of the solution through the needle tip when the concentration of perillaldehyde in the solutions exceeds a critical value, in which uniform nanofibers are formed [32].

The porosity of the nanofiber films was shown in Figure 3. With the increasing ratio of perillaldehyde, the porosity increased and then decreased. The change in the porosity of G/Z/P nanofiber films could be explained by the hydrogen bonding between the perillaldehyde, gelatin, and zein molecules. At the weight ratio of 5:1:0.04 (G/Z/P), the interaction between perillaldehyde, gelatin, and zein was intense, resulting in the smallest density of nanofibers. At the weight ratio of 5:1:0.02 (G/Z/P), the porosity of the nanofiber membrane (76.10%) was higher than the others, which was not significantly different from the G/Z group (74.33%). Porosity is considered a critical parameter for cellular infiltration and bacteriostasis. The porosities of the G/Z/P (5:1:0.02) nanofiber film were within a suitable range (60–90%) for cellular proliferation [12], which indicated its promising application in antimicrobial packaging materials.

### 3.3. FTIR Analysis

FTIR analysis was used to investigate the functional groups and molecular interactions, and intermolecular interactions among the components within the nanofibers. The FTIR spectra of polymer monomers and nanofiber films were shown in Figure 4. A broad absorption at around 3275 cm^−1^ (N–H stretching vibration) was observed in gelatin, and the peaks near 3068 and 2957 cm^−1^ reflect the O–H and C–H stretching vibrations [33,34]. Three peaks at around 1638 (amide I, C = O and C–N stretching vibrations), 1535 (amide II, N–H bending and C–H stretching vibration), and 1235 cm^−1^ (amide III) are the characteristic bands of gelatin (Figure 4A,B) [33,34]. The absorption bands at 1448, 1394, 1334, and 1080 cm^−1^ are attributed to the N–H bending, C–N stretching combination, C–N stretching, and C–H deformation of the methyl group, respectively [20]. After cross-linking of gelatin and zein, their characteristic peaks also appeared on the G/Z nanofiber film. However, the intensities of the absorption peaks at 3276 and 1628 cm^−1^ in the G/Z nanofiber film were stronger than those in the gelatin and zein powders. This result indicated that hydrogen bonds were formed between the polymers during the process of preparing electrospun films [19,35,36].

As shown in Figure 4A,B, the absorption bands at around 2930 (aliphatic C–H stretching), 1670 (C = O stretching vibration), 1640 (aromatic C = C stretching, benzene ring), and 1150 cm^−1^ (C–H deformation) are the characteristic bands of perillaldehyde [37]. The spectra of the G/Z/P composite nanofiber samples (Figure 4C,D) reveal that the introduction of perillaldehyde resulted in noticeably stronger absorptions at 3276 (N–H stretching vibration), 1628 (C = O stretching vibrations), and 1531 cm^−1^ (N–H bending). The peaks of G/Z/P (5:1:0.01) and G/Z/P (5:1:0.02) were stronger than those of the other G/Z/P nanofiber films and close to the absorption peak intensity of the G/Z nanofiber membrane. These results indicated that gelatin, zein, and perillaldehyde interacted through hydrogen bonds.

### 3.4. XRD Analysis

The XRD spectra of polymer monomers and the nanofiber films were shown in Figure 5. The structure of α-helix plays a crucial role in the crystalline structures of gelatin and zein [20,38]. Zein has two obvious peaks at around 9.3° and 19.5°, whereas gelatin has a narrow peak at 8.8° and a broad diffraction peak at 20.2°. The G/Z nanofiber film had two narrow peaks at 7.92° and 19.4° and a broad diffraction peak at 28.77°. Ki et al. [39] reported that the low crystallinity of most G/Z nanofibers may be caused by the acid degradation of their ordered structure. Moreover, the crystallization of polymers could also be hindered during the electrospinning process [40]. All the G/Z/P nanofiber films have two diffraction peaks with different intensities at around 20° and 28°. The peaks are different from those of the G/Z nanofiber film, which showed a peak in the range of 20°–30°. This result indicated that gelatin, zein, and perillaldehyde have good compatibility and interaction in the nanofiber films. As shown in Figure 5, G/Z/P (5:1:0.01) and G/Z/P (5:1:0.02) have a broad peak with high intensity at around 20°, which indicated strong intermolecular and intramolecular hydrogen bonding between polymers.

### 3.5. Thermal Stability

The thermal stability of the nanofiber films was investigated by DSC and TGA, and the result is shown in Figure 6. The characteristic endothermic peaks of the DSC curve are termed as glass transition temperature (*T_g_*) and melting point (*T_m_*) [41,42]. All the nanofiber films exhibited three peaks. The first peak indicates that the polymer (including the amorphous part of the crystalline polymer) changed from the glass state to the highly elastic state [42]. The *T_g_* values of G/Z/P (5:1:0.0025), G/Z/P (5:1:0.005), G/Z/P (5:1:0.01), G/Z/P (5:1:0.02) and G/Z/P (5:1:0.04) were 60.96, 64.87, 65.73, 68.09, and 66.14 °C, respectively. The *T_g_* value of G/Z/P (5:1:0.02) was the highest; thus, the G/Z/P (5:1:0.02) film had the best heat resistance among the films. The second peak represents the energy required for the evaporation of water. The third in the range of 200–230 °C is ascribed to the thermal degradation of the G/Z and G/Z/P films. The *T_m_* values of G/Z/P (5:1:0.0025), G/Z/P (5:1:0.005), G/Z/P (5:1:0.01), G/Z/P (5:1:0.02) and G/Z/P (5:1:0.04) were 215.73, 215.99, 216.07, 216.15, and 215.16 °C, respectively, as shown in Figure 6A, which indicated that the addition of perillaldehyde could improve the thermal stability of nanofiber films.

The thermal degradation of biopolymers involves the degradation of the inner covalent bonds of polymer monomers and film network structure [43]. The TGA curves of the G/Z/P nanofiber films have two zones of weight loss in approximately 55–120 and 214–450 °C, which were caused by the evaporation of moisture and the thermal degradation of protein and perillaldehyde in the film, respectively. Corradini et al. [44] and Deng et al. [20] stated that the decomposition temperatures of gelatin and zein are approximately 220 and 280 °C, respectively. The residual amount of the G/Z/P film at high temperature decreased with the increase in perillaldehyde under the same weight as the test sample, as shown in Figure 6B; hence, perillaldehyde is less stable than gelatin and zein. This result is similar to the result reported by Karim et al. [9].

### 3.6. Water Contact Angle Analysis

The water contact angle results of the nanofiber films were shown in Figure 7. The higher proportion of hydroxyl groups in the structure leads to the superhydrophilic property of the gelatin [45]. As reported by Deng et al. [20], drops of water droplets on the pure gelatin film were absorbed immediately, and convex shape could not be kept. In the present study, drops of deionized water on the G/Z nanofibers could form a convex shape, and its contact angle was 54.6°. The relatively high proportion of hydrophobic groups in zein and the hydrogen bonds formed by the hydrophilic groups between gelatin and zein resulted in the hydrophobic surface of the G/Z nanofiber film [20]. As shown in Figure 7, the water contact angles of G/Z/P (5:1:0.0025), G/Z/P (5:1:0.005), G/Z/P (5:1:0.01), G/Z/P (5:1:0.02) and G/Z/P (5:1:0.04) were 67.3°, 68.7°, 75.8°, 88.4°, and 141.5°, respectively. The water contact angle of the G/Z/P films gradually increased with the increase in perillaldehyde. This result indicates that the G/Z/P nanofibers gradually changed from hydrophilic to hydrophobic as the ratio of perillaldehyde increases.

In the electrospinning process, a radical protein gradient is formed inside the droplet and moves inside toward the center of the droplet within a very short time before the solvent in the ejected droplet is completely evaporated [45]. At this moment, the hydrophobic groups of protein molecules are forced outside by the relatively non-polar air side [20]. Simultaneously, the addition of perillaldehyde increased the ratio of hydrophobic groups and the hydrogen bond interaction between gelatin, zein, and perillaldehyde as indicated by DSC and FTIR analysis. These changes result in a more hydrophobic surface.

### 3.7. Antibacterial Activity

Perillaldehyde is the main antibacterial ingredient of perilla essential oil and is a food additive allowed to be used in China [24,46]. The effects of different G/Z/P nanofiber films in inactivating *S. aureus* and *S. enteritidis* were evaluated by the plate counting method. As shown in Figure 8, the bacterial counts for *S. aureus* and *S. enteritidis* after treatment with G/Z/P (5:1:0.02) and G/Z/P (5:1:0.04) were significantly lower than those of other films when the treatment time was up to 4 h (*p* < 0.05). The viable bacterial counts of *S. aureus* and *S. enteritidis* after 4 h of treatment with G/Z/P (5:1:0.02) and G/Z/P (5:1:0.04) had no significant difference (*p* > 0.05). The results indicated that the nanofiber films of G/Z/P (5:1:0.02) and G/Z/P (5:1:0.04) had a better antibacterial effect than the other films.

### 3.8. Antibacterial Activity of Nanofiber Film on Chicken Breasts

The existence and growth of spoilage bacteria are one of the important reasons for the spoilage of meat products [47]. The above results showed that the G/Z/P (5:1:0.02) among the five nanofiber films had the best nanofiber characteristic and better antibacterial activity. The G/Z/P (5:1:0.02) can be used as a better food package cling film to control meat spoilage. The changes of total viable counts in the chicken breasts during storage were shown in Figure 9A. The initial total viable counts in the chicken breast were 3.43 log CFU/g, and it gradually increased during the storage. The increase rate of the control samples was significantly faster than that of the G/Z/P (5:1:0.02) (*p* < 0.05). After nine days of storage, the total viable counts of the control sample was increased to 7.73 log CFU/g, while the total viable counts of the G/Z/P (5:1:0.02)-packaged sample was only 4.72 log CFU/g. Chotimarkorn proposed that the upper total viable count limit for fresh meat is in the range of 6 to 7 log (CFU/g) [48]. At 12 days, the total viable counts in the G/Z/P (5:1:0.02) group was 5.95 log CFU/g, which was close to spoilage, while the meat of the CO had long been spoiled. Therefore, the antibacterial properties of G/Z/P (5:1:0.02) nanofiber film significantly inhibit the growth activity of microorganisms and delay the deterioration of chicken.

TVB-N is one of the important indicators for evaluating the freshness of meat products, which is mainly composed of basic nitrogenous substances, such as ammonia and amines [49]. The changes of TVB-N values in chicken breasts during storage were shown in Figure 9B. The TVB-N values significantly increased as the storage time prolonged (*p* < 0.05), which may be due to the presence of spoilage microorganisms and the decomposition of meat protein [50]. Compared with the control samples, the TVBN values of chicken breasts packaged with G/Z/P (5:1:0.02) film increased slowly during storage (Figure 9B). After 12 days of storage, the TVB-N value of the G/Z/P (5:1:0.02)-packaged samples was 15.55 mg/100g, which was significantly lower than the control of 41.77 mg/100g (*p* < 0.05). The results indicated that the G/Z/P (5:1:0.02) film could effectively inhibit the increase of TVB-N, which was due to the antibacterial property of perillaldehyde. Xin et al. [16] also reported similar findings that nanofiber films loaded with curcumin and chitosan could effectively inhibit the increase of TVB-N in Schizothorax prenati Fillets. In this study, G/Z/P (5:1:0.02) film can effectively inhibit the growth activity of spoilage microorganisms in chicken meat and extend the preservation time of chilled chicken breast.

## 4. Conclusions

G/Z/P nanofiber films were successfully prepared by electrospinning technique using three food grade ingredients of gelatin, zein, and perillaldehyde. These three ingredients interacted strongly via hydrogen bonding, and G/Z/P (5:1:0.02) had the best nanofiber structure in these G/Z/P films. The addition of perillaldehyde enhanced the thermal stability, density, and hydrophobicity of G/Z/P nanofiber films. The G/Z/P (5:1:0.02) and G/Z/P (5:1:0.04) nanofiber films effectively inactivate two foodborne pathogens of S. aureus and S. enteritidis. The chicken breast packaged with G/Z/P (5:1:0.02) had obviously longer shelf time than the control. The comprehensive observation of morphology, stability, and antibacterial activity of the different prepared films indicated that G/Z/P (5:1:0.02) has a potential application in antibacterial food packaging.

## Figures and Tables

**Figure 1 foods-10-01277-f001:**
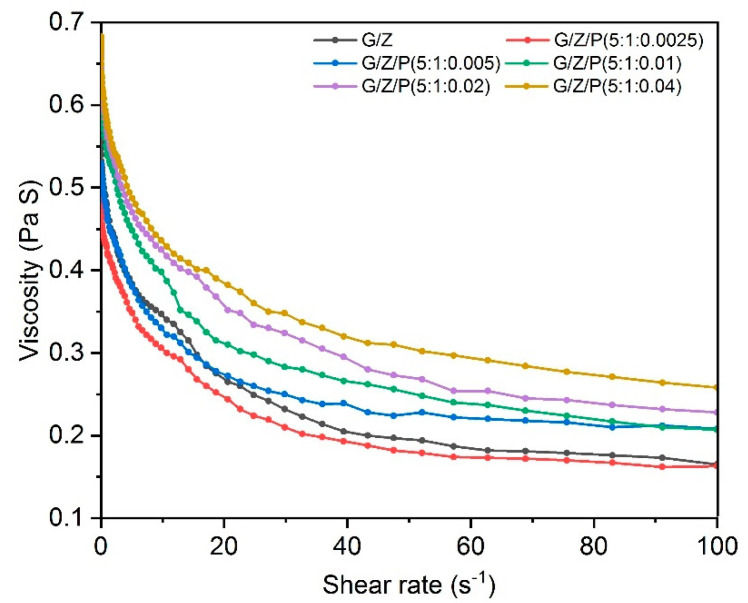
The viscosity of gelatin/zein solutions loaded with different concentrations of perillaldehyde.

**Figure 2 foods-10-01277-f002:**
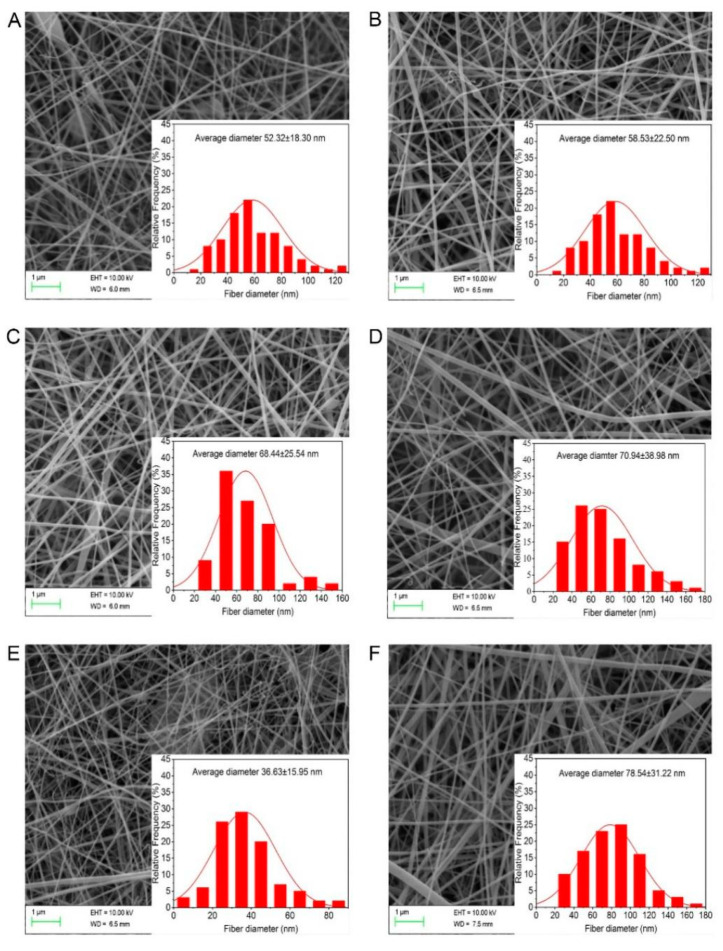
SEM images and diameter distributions of the gelatin/zein/perillaldehyde nanofiber films. (**A**) G/Z/P (5:1:0.0025), (**B**) G/Z/P (5:1:0.005), (**C**) G/Z/P (5:1:0.01), (**D**) G/Z/P (5:1:0.02), (**E**) G/Z/P (5:1:0.04), (**F**) G/Z.

**Figure 3 foods-10-01277-f003:**
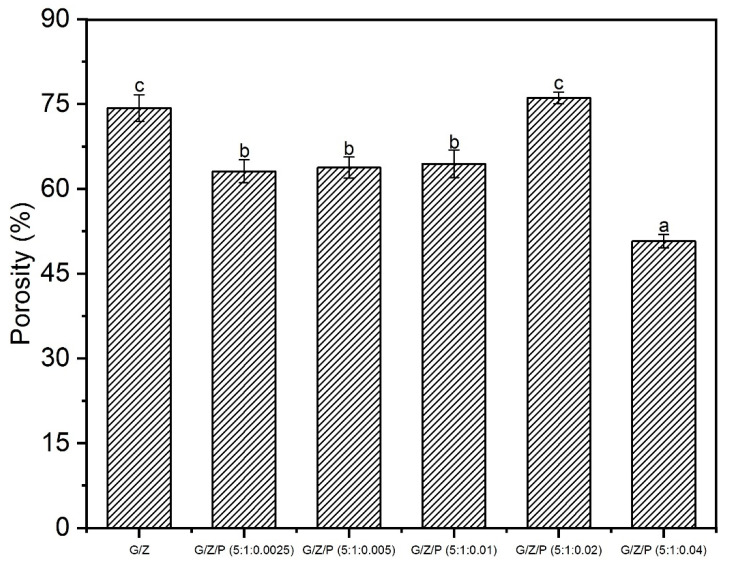
The porosity of the gelatin/zein/perillaldehyde nanofiber films. The values are given as the mean ± SD, and different letters in the same line mean the significant difference (*p* < 0.05).

**Figure 4 foods-10-01277-f004:**
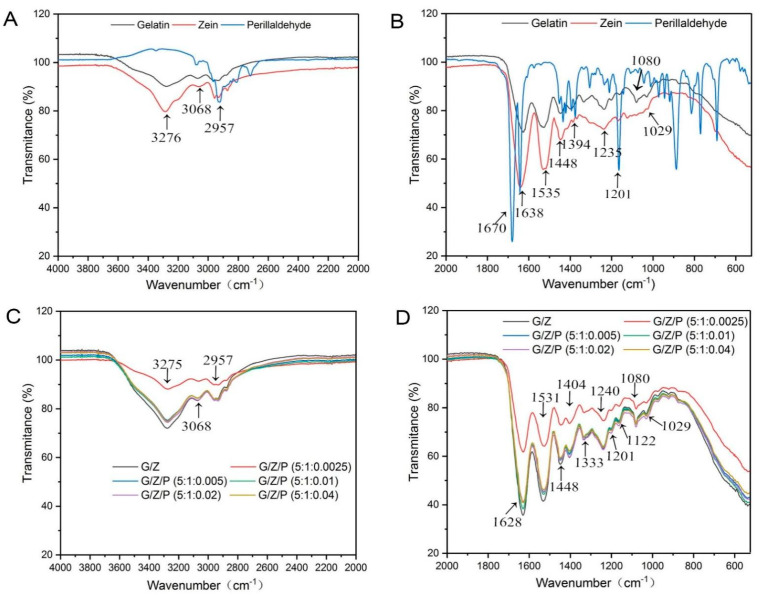
FTIR spectra of polymer monomers of gelatin, zein and perillaldehyde (**A**, **B**) and the gelatin/zein/perillaldehyde nanofiber films (**C**, **D**).

**Figure 5 foods-10-01277-f005:**
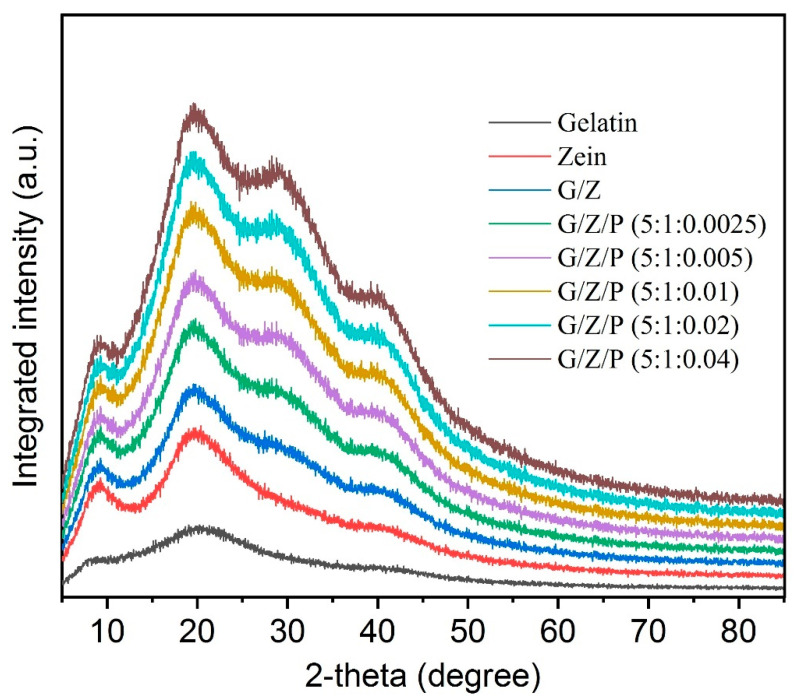
X-ray profiles of the gelatin/zein/perillaldehyde nanofiber films.

**Figure 6 foods-10-01277-f006:**
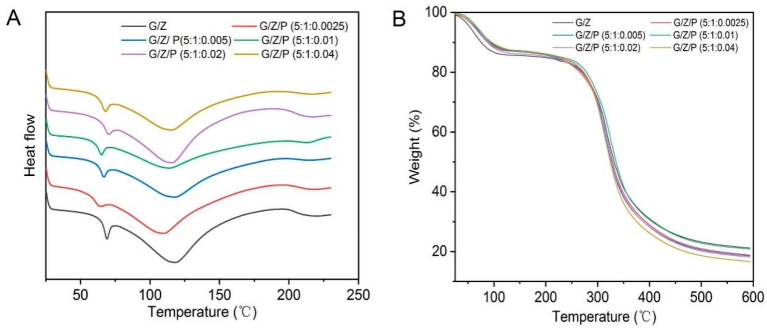
DSC curves (**A**) and TGA thermogram curves (**B**) of the gelatin/zein/perillaldehyde nanofibers.

**Figure 7 foods-10-01277-f007:**
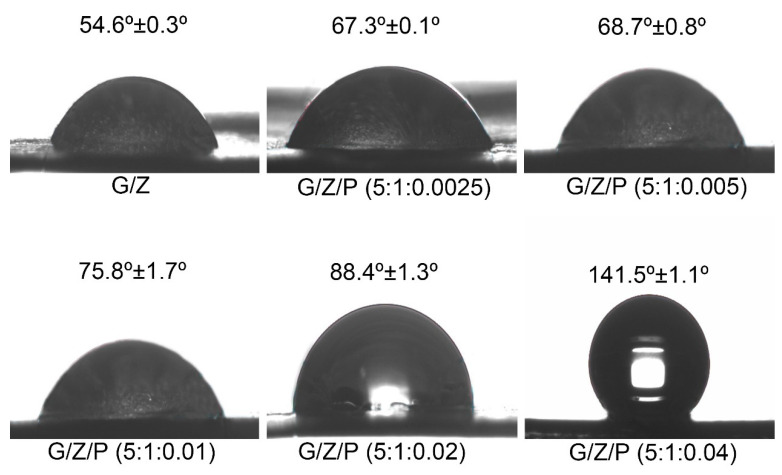
The water contact angle of the gelatin/zein/perillaldehyde nanofiber films.

**Figure 8 foods-10-01277-f008:**
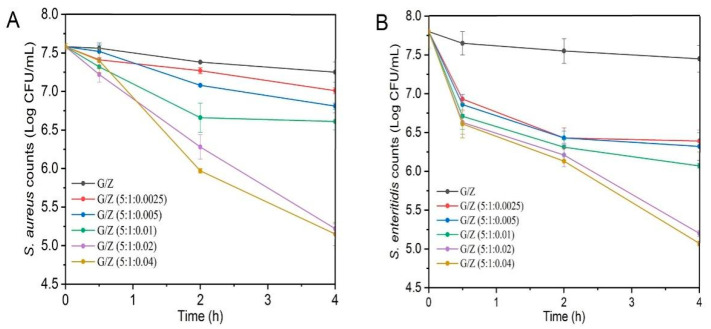
Inactivation of S. *aureus* and S. *enteritidis* by the gelatin/zein/perillaldehyde nanofiber films.

**Figure 9 foods-10-01277-f009:**
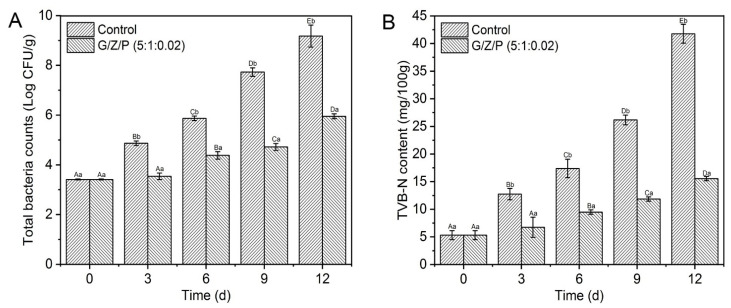
Impact of G/Z/P (5:1:0.02) nanofiber film on total viable counts (**A**) and TVB-N (**B**) of chicken breasts during storage. ^a,b^ Mean values in the bar, corresponding to the same days of storage, not followed by common lowercase letters, differ significantly (*p* < 0.05). ^A,B,C,D,E^ Mean values in the bar, corresponding to the same group, not followed by common uppercase letter differ significantly (*p* < 0.05).

## Data Availability

All data not application.

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
