# Peer review of "Fabrication and Characterization of Gelatin/Zein Nanofiber Films Loading Perillaldehyde for the Preservation of Chilled Chicken"

_foods, 2021, doi:10.3390/foods10061277_

Round 1
Reviewer 1 Report
The article entitled „Fabrication and characterization of gelatin/zein nanofiber films loading perillaldehyde for the preservation of chilled chicken” by Wang and co-workers was subjected to be published in FOODS journal. In this research article authors described process of obtaining new nanofiber films composed of gelatin, zeine and different concentrations of perillaldehide, the substance extracted from perilla essential oil.
The research was conducted properly, using a broad spectrum of instrumental techniques necessary to study physical properties of obtained material. A studies of antibacterial activity of nonofiber films were also performed. Although the technique of nanofiber production and its physical characterisation is clearly described, the results of antibacterial activity of these materials is a bit confusing. Only two bacterial species were chosen as a model for all experiments: Staphylococcus aureus and Salmonella enteritidis. Both of them could be recognized as an example of the model Gram-positive and Gram-negative bacterium. But there are others pathogenic strains most frequently isolated from chilled meat, for example Listeria monocytogenes – a very dangerous intracellular pathogen, the presence of which in raw chicken meat causes its disqualification and withdrawal from the sale. It would be very interesting to see the results of experiments concerning antibacterial activity of the new nanofiber films also at this bacteria species.
Among this, I would like to point to the next drawback of this manuscript: at the Figure 8 the inhibition zone diameter is hardly visible. All the Petri dishes are covered by really abundantly growing bacteria and only in the immediate vicinity of nanofiber discs there are visible very small, maybe 1 mm inhibition zone. So, the effect of growth inhibition of both tested bacteria is very low. Much better looks the results of the next experiments, concerning the inactivation of pathogen cells. Authors should comment on the diffusion plate assay, or improve the picture, or – if is not possible to improve just remove this experiment from the manuscript.
Minor remarks:
- Page 3, line 104 and 112; it is: 25 x 25 cm2, and 3 x cm2. In my opinion should be: 25 cm x 25 cm, and 3 cm x 3 cm.
- Page 6, Figure 2: This figure should be magnified, the scale and figures descriptions are hardly to read and invisible, especially at the inserts.
- Page 8, Figure 4: The FT-IR spectra should be magnified at least twice. All of them are illegible.
- Page 13, Figure 10. Each bar has two letters at the top, but there is no explanation of significance in the Figure description. The overview of the statistical analysis is impossible to do, because the letters are too small and hardly visible. Anyway, the bar size is proper.
- References: please pay attention for the italic style in the name of bacteria species (e.g. positions 24, 28, 49, 50).
Author Response
24nd May, 2021
Dear Mr. Paul Qi and Reviewer,
We would like to thank you and the Reviewers' for their constructive comments on our Manuscript ID foods-1194399. We have amended the manuscript based on the comments and highlighted our major or small changes in the text using red type and provided line numbers for those changes. Please see the revised manuscript and our point-by-point responses. We look forward to your further comments.
With best wishes,
Yours sincerely,
Debao Wang
We would like to express our sincere thanks to the reviewers for the constructive and positive comments.
Reviewer(s)' Comments to Author:
Reviewer: 1
Among this, I would like to point to the next drawback of this manuscript: at the Figure 8 the inhibition zone diameter is hardly visible. All the Petri dishes are covered by really abundantly growing bacteria and only in the immediate vicinity of nanofiber discs there are visible very small, maybe 1 mm inhibition zone. So, the effect of growth inhibition of both tested bacteria is very low. Much better looks the results of the next experiments, concerning the inactivation of pathogen cells. Authors should comment on the diffusion plate assay, or improve the picture, or – if is not possible to improve just remove this experiment from the manuscript.
Replies to Reviewer 1
1.Authors should comment on the diffusion plate assay, or improve the picture, or – if is not possible to improve just remove this experiment from the manuscript.
Response: Thank you for your advice, we have removed that experiment from the manuscript.
2.Page 3, line 104 and 112; it is: 25 x 25 cm2, and 3 x cm2. In my opinion should be: 25 cm x 25 cm, and 3 cm x 3 cm.
Response: Thank you for your advice, we have made modifications according to your suggestions, and the modifications have been marked in red.
3.Page 6, Figure 2: This figure should be magnified, the scale and figures descriptions are hardly to read and invisible, especially at the inserts.
Response: Thank you for your serious advice. the scale and figures descriptions of figure 2 has been magnified and the modified image has been re-inserted.
4.Page 8, Figure 4: The FT-IR spectra should be magnified at least twice. All of them are illegible.

Reviewer 2 Report
Dear Authors,
Your article is very interesting and I was pleased to review it.
- What is the main question addressed by the research?
The main question was whether the resulting films are antibacterial and whether they are suitable for the food industry.
-Is it relevant and interesting?
The subject of this article is important and relevant from the point of view of the food and packaging industry. Despite the large amount of work on this topic, new solutions are still being sought.
- How original is the topic?
The subject is very original due to the growing demand for packaging containing natural biocides. Perillaldehyde is a natural antibacterial agent extracted from perilla essential oil.
- What does it add to the subject area compared with other published material?
Particular analyzes have shown obtaining a product with the desired physico-chemical and biological properties. The article introduces a broad description of the new product.
- Is the paper well written? Is the text clear and easy to read?
The text is clear easy to read.
- Are the conclusions consistent with the evidence and arguments
presented?
The conclusions are in line with the results obtained.
The comprehensive observation of morphology, 403 stability, and antibacterial activity of the different prepared films indicated that G/Z/P 404 (5:1:0.02) has a potential application in antibacterial food packaging.
- Do they address the main question posed?
The conclusions are clear and answer the main question.
The product obtained with the desired parameters, can be used in the food industry.
The chicken breast packaged with G/Z/P (5:1:0.02) had obviously longer shelf time than the control.
Author Response
24nd May, 2021
Dear Mr. Paul Qi and Reviewer,
We would like to thank you and the Reviewers' for their constructive comments on our Manuscript ID foods-1194399. We have amended the manuscript based on the comments and highlighted our major or small changes in the text using red type and provided line numbers for those changes. Please see the revised manuscript and our point-by-point responses. We look forward to your further comments.
With best wishes,
Yours sincerely,
Debao Wang
Reviewer: 2
Dear Authors,
Your article is very interesting and I was pleased to review it.
Replies to Reviewer 2
1.What is the main question addressed by the research?
Response: Thank you for your question. The main question was whether the resulting films are antibacterial and whether they are suitable for the food industry.
2.Is it relevant and interesting?
Response: The subject of this article is important and relevant from the point of view of the food and packaging industry. Despite the large amount of work on this topic, new solutions are still being sought.
3.How original is the topic?
Response: Thank you for your precious question. The subject is very original due to the growing demand for packaging containing natural antibacterial material. Perillaldehyde is a natural antibacterial agent extracted from perilla essential oil.
4.What does it add to the subject area compared with other published material?
Response: Thank you for your precious question. Particular analyzes have shown obtaining a product with the desired physico-chemical and biological properties. The article introduces a broad description of the new product.

This manuscript is a resubmission of an earlier submission. The following is a list of the peer review reports and author responses from that submission.